# Recent Progress on Optical Tomographic Technology for Measurements and Inspections of Film Structures

**DOI:** 10.3390/mi13071074

**Published:** 2022-07-07

**Authors:** Ki-Nam Joo, Hyo-Mi Park

**Affiliations:** 3D Optical Metrology Laboratory, Department of Photonic Engineering, Chosun University, 309 Pilmun-daero, Dong-gu, Gwangju 61452, Korea; hyomi0425@gmail.com

**Keywords:** film metrology, tomographic measurement, ellipsometry, reflectometry, interferometry, confocal microscopy, hybrid film metrology, machine learning

## Abstract

In this review, we present the recent progress on film metrology focused on the advanced and novel technologies during the last two decades. This review consists of various technologies and their measurement schemes to provide the inspiration for understanding each of the measurement principles and applications. In the technology and analysis section, several optical techniques used in film metrology are introduced and described with their benefits and limitations. The temporal, spatial and snapshot measurement schemes of optical film metrology are introduced in the measurement scheme section, and finally, the prospect on optical film metrology will be provided and discussed with the technology trend.

## 1. Introduction

In the era of fourth industry, the importance of film structures has been drastically emphasized the development of more intelligent and smarter high-end devices [1,2,3,4,5]. In the semiconductor industry, the level of integration in memory chips has rapidly increased according to the demands of high-performance electronic products, and the patterns on system semiconductors have been more complex as the desired functionalities increase [6,7]. In this case, the semiconductors should be stacked, and the circuits need to be designed as 3D structures in addition to the multi-layered film depositions on the wafer in a single chip [8]. In the meantime, the display products based on OLED and μ-LED for virtual reality (VR), augmented reality (AR), and flexible screens are also required to be manufactured as complicated film structures for high contrast, brightness and fast response [9,10].

In the manufacturing process of these film structures, most important issue is to confirm and stabilize the optimal process conditions to reduce the defects on the products, and the characterizations of the film structures are essential. The uniformity of the materials and their dimensions such as film thicknesses should be inspected and measured to determine the manufacturing process parameters. Therefore, this measurement and inspection (MI) of the film layers can be involved as the in-line process during manufacturing the devices [11,12], or it can be implemented at the final stage of manufacturing [13]. Regardless of the application point, however, every MI technology needs high precision and fast measurement capability to characterize multi-layered films.

Meanwhile, the film metrology has been applied to distinguish the biological layers in eyes [14], teeth [15] and bones [16] as optical tomographic tools in biomedical fields. By adopting interferometric principles, optical coherence tomography (OCT) has been increasingly developed and commercialized in the biomedical area. Furthermore, the biomedical devices such as stents, implants, wires and catheters need the surface coating to protect the devices from corrosion as well as infection from disease, and the coated film should be carefully inspected to avoid unexpected hazards [17,18].

Typically, a scanning electron microscopy (SEM) and a transmission electron microscopy (TEM) have been widely used to inspect the metallic, organic and inorganic film structures with high precision [19,20]. The SEM can produce images of a sample by scanning the surface with a focused beam of electrons. The electrons interact with atoms in the sample, producing various signals that contain information about the surface topography and composition of the sample [19]. In transmission electron microscopy (TEM), an image is formed from the interaction of the electrons with the sample as the beam is transmitted through the specimen [20]. They are capable of imaging with the significantly higher resolution than light microscopes, owing to the smaller de Broglie wavelength of electrons. However, the sample should be well prepared, which means the sample should be destroyed to increase their electrical conductivity and to stabilize it. In addition, they should be typically operated in vacuum environment, which makes it difficult to be used in manufacturing processes and biomedical applications.

On the other hand, optical techniques are attractive in the industrial and biomedical fields because of their non-destructive, highly sensitive and precise measurement characteristics, and manifold research studies have been reported, which led to the commercialization. Reflectometry and ellipsometry are the representative optical measurement methods for film metrology. The reflected light from a sample is collected and analyzed in the spectral domain with the theoretical model of film layers. Optical interferometry has been commonly used for the topographic measurements of samples, but it can be used in film metrology as a tomographic tool with the aids of film theory. Confocal microcopy is also one of the optical tomographic measurement techniques in biomedical fields with the acquisition of the best focal positions corresponding to the film interfaces.

In this review, we summarize the recent progress on the optical tomographic technologies for film metrology. Because the traditional optical techniques such as reflectometry, ellipsometry, interferometry and confocal microscopy as the general point of view are well described in lots of textbooks related to optical metrology, we focus on the advanced and novel technologies during last two decades. This review consists of two categories: (1) technology and analysis and (2) measurement scheme. In the technology and analysis section, several optical techniques used in film metrology are introduced and described with their benefits and limitations. The temporal, spatial and snapshot measurement schemes of optical film metrology are introduced in the measurement scheme section, and finally, the prospect on optical film metrology will be provided and discussed with the technology trend.

It is noted that the scope of this review is put on the recent developments of the measurement technology and system. Therefore, the research works on the applications of film metrology to purely characterize material properties are out of the scope, but some of them are inevitably involved in this review. Moreover, this review cannot practically contain all research works related to optical film metrology, but we believe most of the heavily cited research studies are considered and discussed. The technical terminologies used in this review are determined as a principle-based point of view for the unity and coherence of this review because lots of various technologies are introduced. For example, a ‘time-domain OCT (TD-OCT)’ is expressed as a ‘low-coherence scanning interferometry (LCSI)’ and so on.

## 2. Technology and Analysis in Optical Film Metrology

Figure 1 shows the category of film metrology in the technical point of view. Electron microscopy such as SEM and TEM is a general tool to observe the surface and inside of a film layer with high lateral and vertical resolution used in industrial and biomedical fields. However, the specimen should be well defined, which means it needs to be destroyed and well prepared with the additional process such as metallic coatings. Moreover, its applicability is very limited because of the vacuum environment where it is installed.

In non-destructive techniques of film metrology, X-ray tomography and ultrasound techniques are very useful for the characterization of film structures, which especially contain opaque or metallic film layers. Terahertz imaging techniques are also potential approaches to detect structural defects in ceramic and composite materials and imaging the physical structure of paintings. On the other hand, optical film metrology is a well-established technology to measure and inspect film structures in semiconductor and display products as well as observe biomedical samples. It is a non-contact and non-destructive analyzing technique, which can be conveniently used in wide applications, with high reliability and precision, although the film layers should be optically transparent. In addition, it does not need any special environmental conditions such as vacuum.

Optical film metrology can be typically categorized as ellipsometry, reflectometry, interferometry and confocal microscopy. Ellipsometry and reflectometry have been traditionally used for measuring thin and thick film layers, while interferometry and confocal microscopy are suitable for measuring relatively thick film layers or plate thicknesses. To simultaneously obtain the surface and thickness profiles of the specimen, interferometry and confocal microscopy are combined with reflectometry and ellipsometry as hybrid approaches. In recent years, a machine learning technique has been adopted to analyze more complicated film structures. It is noted that there is no critical boundary between thick and thin film thicknesses, but a 100 nm thickness was used as a criterion to distinguish them in this review.

### 2.1. Ellipsometry

Ellipsometry has been a standard measurement tool to measure thin film layers and their refractive indices in industry and science fields. Since Drude firstly built an ellipsometer, various configurations have been proposed and commercialized. Ellipsometry fundamentally measures the changes in the state of polarization of light upon reflection from a surface of the specimen. Recently, most research works have been focused on spectroscopic ellipsometry (SE) and its applications because the ellipsometric angles (*Ψ*, Δ) in the broad spectral range of the source obtained by SE can produce measurement results that are more reliable and accurate [21,22,23,24,25,26,27,28,29,30,31,32]. However, approaches to improve the measurement accuracy to decouple the film thickness and the refractive index in single-wavelength ellipsometry have been continuously attempted because of its simple cost-effective configurations [33,34,35]. The phase modulation by a photoelastic modulator was used in single wavelength and single angle ellipsometry [33] to identify the specific conditions to decouple the film thickness and the refractive index of a single-layered film structure. Based on the relationship between Bessel functions obtained by a Fourier transformation of the harmonic phase modulations, the ellipsometric angles were extracted, and the specific conditions for various specimen were provided. As another attempt, the alternative analytic model was proposed based on Drude’s approach to calculate the complex reflection coefficients from electromagnetic theory. Opposed to the traditional way, a series expansion of the ellipsometric ratio (*ρ*) to the second order of the film thickness relative to the wavelength was adopted to find the linear relationship between Re[*ρ*] and (Im[*ρ*])^2^, of which the slope only depends on the refractive index of the film. As the result, the proposed technique enabled to determine simultaneously both thickness and refractive index of ultra-thin films down to 5 nm thickness with the aid of an additional reference measurement [35].

Spectroscopic imaging ellipsometry (SIE) was initially proposed to improve the lateral resolution with the aid of an imaging technique, but SIE has been proposed to obtain the film thickness profiles at the large measurement areas [27,28,29,30,31,32] in recent years. By adopting a low magnification microscope objective with an imaging spectrometer as shown in Figure 2A, a line profile of film thicknesses up to 16.2 mm was obtained by the single operation, and the 3D film thickness profile of the specimen was reconstructed with the lateral scanning of the specimen, as shown in Figure 2B [31].

In addition, a non-collimated beam was used to illuminate the specimen in SIE [36], and the 3D film thickness profile was obtained at once in PSA-type ellipsometry. In this research work, a wide-angle imaging ellipsometric configurations as shown in Figure 3A, respectively, were proposed, and the thickness profiles of various SiO_2_ film layers were reconstructed as shown in Figure 3B. However, the lateral resolution was determined by the angular resolution limited by the exit pupil of the system, and it was 0.5 mm.

In the meantime, several research studies related to total internal reflection ellipsometry (TIRE) were conducted to significantly enhance the thin film sensitivity [24,25,28]. Especially, TIRE has the benefit of monitoring processes on a surface in an opaque solution. TIRE was configured as a spectroscopic ellipsometer with the Kretchmann configuration of surface plasmon resonance (SPR), as shown in Figure 4A [24]. At the critical angle, the energy of the light incident to the thin metal film/dielectric interface is partially transferred to plasmons, which leads to reducing the reflectivity. Then, the position of a sharp peak on the reflectivity, i.e., SPR, its amplitude and width depend on the optical parameters of metal films. Moreover, the extremely high sensitivity of the SPR peak position is determined by the thickness and/or refractive index of the film. The combination of SE and SPR, therefore, in TIRE has the benefit for several applications in both gaseous and liquid media as well as for thin film characterization, as shown in Figure 4B [24]. However, the use of a prism for the total internal reflection restricts various applications of TIRE for general purposes.

On the other hand, non-traditional ellipsometers were also introduced to improve the fundamental limitation of ellipsometry. A multiple incidence medium (MIM) technique similar to multiple incidence angle (MIA) was invented for measuring ultrathin film thicknesses [34]. In the film regime less than 10 nm thickness, the corresponding changes of *Ψ* become extremely small, and the measurement system requires the high precision of the measurement result. In the MIM ellipsometer, two measurement results of Δ were carried out under variation of the ambient refractive index. In the research work, the thickness and the refractive index of a single layered film were simultaneously determined by single wavelength ellipsometry in different ambient media, although a special container to change the medium was needed.

As another approach, spatial ellipsometry, which spatially detected the intensity variations instead of rotating optical components, was proposed by using a spatial phase retarder or a polarization pixelated camera, as shown in Figure 5 [37]. The spatially phase-retarded spectroscopic ellipsometry used a pseudo depolarizer, which consisted of a rotating liquid crystal (LC) periodic array as the spatial phase retarder, and it was able to collect all information necessary to characterize film structures with a single image acquisition with the benefit of real-time measurements. Spatial, spectral and polarization information of the film structure was able to be acquired at the frame rate of the camera without any mechanical rotations of polarizing optical components with the aid of using a polarization-pixelated CMOS camera and an imaging spectrometer. Although this spatial technique in ellipsometry provides the possibility of snapshot measurement, further achievement regarding precision and accuracy is necessary.

### 2.2. Reflectometry

Reflectometry has been conveniently used in film metrology because of its simple configuration and vertical measurement probe as opposed to ellipsometry. Especially, the spectroscopic reflectometry has been widely used in various industrial and biomedical applications such as dielectric and liquid film layers [38,39,40,41,42,43,44,45,46,47,48,49,50,51]. Some research works were related to imaging reflectometry to reconstruct the volumetric film structures [40,41,42,43,44,45,46]. Using a monochromator [41,42,44], acousto-optic tunable filter [40] and various tunable bandpass filters [46], each spectral reflectance corresponding to the surface was obtained by an imaging device (we show an example of an imaging spectroscopic reflectometer in Figure 6A), and the spectral reflectance at each pixel was compared with the theoretical model of the film structure, as shown in Figure 6B [40].

However, the typical reflectometry is suitable for thick film thickness or single-layered film thickness measurements because of its relatively lower sensitivity. At the absence of preliminary knowledge of the medium, reflectometry is not capable of separating the film thickness and refractive index of the film layer. To overcome this limitation, several types of reflectometry have been reported. Ellipsometric reflectometry (the authors named ‘optical fixed-angle reflectometry’) allowed the ellipsometric configuration to reflectometry, as shown in Figure 7A, to analyze the reflectivity of *p*- and *s*-waves [47]. It was quite similar to ellipsometry, but the theoretical model and the system analysis were different from those of ellipsometry. Using a He-Ne laser, the optical fixed-angle reflectometry was able to determine the change in refractive index and the thickness of the polyelectrolyte film, as shown in Figure 7B. However, most of the research works related to reflectometry over the past two decades have been focused on combined or hybrid techniques with optical low coherence interferometry [48,49,50,51]. It was because of the ability of phase measurement in interferometry, and both techniques are complimentary with each other. The hybrid system for film metrology will be considered in Section 2.5.

### 2.3. Confocal Microscopy

Confocal microscopy is capable of measuring the topographic profile of a specimen based on the detection of light at the best focus. The light reflected off on the surface of the specimen becomes maximal at the best focal position, which can provide the surface height of the specimen. In confocal scanning microscopy (CSM), the specimen is being scanned along the optical axis, and the reflected light is continuously recorded to find the position of the maximum intensity of the light. The CSM can be also used in film metrology to distinguish the film layers by separation of the intensity peak signals [52,53,54,55]. Because of the simple configuration and operation of CSM, it has been applied to measuring the film thicknesses of the industrial specimen such as microbubbles, heat pipes and microfluidic devices.

However, the axial resolution of CSM is strongly dependent on the numerical aperture (N.A.) of an objective lens of the system, and it is quite appropriate for measuring the thicknesses of thick film layers or transparent plates. In addition, it cannot separate the physical thickness and the refractive index of the film because of its fundamental theory based on the refraction. To overcome the limitation, a specific configuration of CSM has been proposed to obtain both of the physical thickness and the refractive index of the media. In the dual-confocal fiber-optic sensor [56], as shown in Figure 8A, a geometrical-ray model as illustrated in Figure 8B was used to obtain the analytical dependence between the thickness and the refractive index of a transparent plate with high accuracy, although the minimum measurable thickness was 1.7 μm.

On the other hand, the recent system research works in confocal microscopy have been focused on the non-scanning type, so-called chromatic confocal microscopy (CCM), to use the chromatic aberration of optical components instead of adopting axial scanning mechanisms [57,58,59]. Corresponding to the chromatic focal positions, each wavelength of the reflected light is detected by a spectrometer, and the relationship between the focal positions and the wavelength shifts enables calculating the surface height of the specimen. The CCM has been used to measure the thickness of a glass plate [57], as shown in Figure 9. However, the minimum measurable thickness by CCM is also limited by the N.A. of the objective lens similar to CSM, and it additionally suffers from the nonlinearity between the chromatic focal positions and wavelength shifts. Moreover, the spectral resolution of the spectrometer restricts the measurement resolutions. The typical resolution and minimum measurable thickness of CSM was in the range of a few micrometers.

### 2.4. Interferometry

Optical interferometry has been widely used as a non-contact, precise and convenient tool to measure the surface profiles, textures and roughness of the specimen. Interferometry is generally categorized as coherent interferometry base on phase shifting techniques and incoherent interferometry using a broadband light source. Although monochromatic phase shifting interferometry (PSI) can be used in film thickness measurements, it needs additional techniques to overcome the 2π-ambiguity and determine the initial position of the target or the interference fringe order [60,61]. Regarding the measurement of continuously varying film layers and its variations [60], PSI was used to evaluate the film thicknesses, and especially, an electronic speckle pattern interferometer (ESPI) was adopted to monitor the temporal variation of film layer thickness [62].

However, most of the research works for film metrology using interferometry have been focused on low-coherence interferometry (LCI) [63,64,65,66,67,68,69,70,71,72,73,74,75,76,77,78,79,80,81,82,83,84,85,86], which has the significant benefit to obtain absolute distances and lengths, and it is suitable for measuring the thicknesses of transparent plates and film layers. LCI can be typically categorized as a scanning interferometer and a spectral interferometer by the operating principle. A low-coherence scanning interferometer (LCSI) or a white light scanning interferometer (WLSI) uses the low temporal coherence characteristics of a broadband light and obtains the localized interference signal, so-called correlogram when the reference and the measurement lengths are close to each other. By adjusting one of them, the peak position of the correlogram represents the height of the target surface. In case of measuring a transparent plate, two correlograms can be detected, and the distance between them indicates the optical thickness of the plate [63]. LCSI has been also used to measure film thicknesses, but it should refer to the mathematical model of the correlogram because the correlograms corresponding to the surface and film layers are not separated any more. The Fourier components of the correlogram include the phase and the amplitude related to the film structure of the specimen including topographic surface height. Based on the theoretical model of the film specimen, the measurement values are compared with the counterparts of the model to minimize the objective function by regression analysis [64,65,66,67,68,69,70,71]. To improve the measurement accuracy and reduce the calculation time, several parameters such as the Fourier phase [66,67], Fourier amplitude [64] and the combined parameter with the Fourier phase and amplitude [66,70,71], for example helical complex field (HCF) function [70], were proposed, and the film thicknesses below sub-μm level were precisely measured. However, LCSI inherently requires the precise scanning mechanism, which limits the real-time measurements.

In a similar fashion, spectrally resolved interferometry (SRI) or dispersive (spectral) interferometry obtains the spectral phase of the interferogram obtained by a spectrometer, and the film thicknesses are extracted via the comparison between the measured phase and its theoretical counterpart by the optimization technique [72,73,74,75,76,77,78,79,80,81,82,83,84,85,86], as shown in Figure 10A [83]. In case of a transparent plate as a specimen, the Fourier transformation is applied, and the plate thickness can be conveniently determined by the Fourier peak positions, as shown in Figure 10B [75]. The benefit of SRI is mainly put on no mechanical moving parts compared to LCSI, but the calculation errors caused by Fourier transformation and filtering functions used in the analysis process can lower the measurement accuracy. In order to reduce the error in the spectral phase, the phase shifting technique can be adopted, and the spectral phase is directly and precisely determined [81,82].

### 2.5. Hybrid Technique

In recent years, the approaches to combine distinct measurement principles have been mostly attempted in film metrology to overcome the limitation of each measurement system [87,88,89,90,91,92,93,94,95,96,97]. Ellipsometry and reflectometry have the ability of measuring film thicknesses, but the topographic height of the film specimen cannot be determined. In order to overcome this limitation, interferometric techniques were introduced in the hybrid measurement system. The combination of the reflectometry and SRI realized the full analysis of a film structure including topographic and tomographic film thicknesses [87,88,89,90,91]. Because the optical configuration of the spectroscopic reflectometry can be modified from that of SRI, the hybrid system was conveniently configured, as shown in Figure 11A. By using polarizing optical components, only the reflected light from the specimen was obtained by the ‘Spectrometer S’ for reflectometry, and the spectral interferogram between the reference and measurement lights was captured by the ‘Spectrometer T’ for SRI. Based on two measurement results of reflectometry and SRI, the topographic and tomographic thickness profiles were simultaneously reconstructed. Other hybrid systems were also proposed: for example, the combination of chromatic confocal microscopy for topographic profiling and reflectometry for film thickness measurements [92], the combination of digital holography for topographic profiling and reflectometry for film thickness measurements [93] and the combination of LCSI for topographic profiling and reflectometry for film thickness measurements [94].

On the other hand, the ellipsometry uses inclined incident light to the specimen, which is significantly different from the configurations of typical interferometry. To avoid conflicting features of two principles, the hybrid system with ellipsometry and interferometry was re-designed as a single measurement system in Figure 11B [95]. By the use of an optical shutter, each measurement was implemented, and the whole measurement results led to the characterization of a film structure.

Furthermore, confocal microscopy and SRI were combined for simultaneously measuring the physical thickness and refractive index of film layers [96,97]. Considering the difference between the theoretical descriptions of two techniques, the physical thickness and the refractive index of a film layer were able to be decoupled, and they were separately measured.

## 3. Measurement Scheme

Because many kinds of optical technologies are used in film metrology, as aforementioned, it is difficult to discuss each detailed measurement scheme in this review. Instead, the overall research trend will be considered in this chapter.

### 3.1. Temporal Acquisition of the Data

For characterization of the film structure, various parameters representing refractive indices and thicknesses of film layers should be determined. In ellipsometry, for example, the ellipsometric angles (*Ψ*, Δ) are necessary, and the phase corresponding to the optical path length needs to be measured in interferometry to extract film thicknesses. Traditionally, the importance of the acquisition was put on the measurement precision and accuracy. Mostly, the temporal acquisition of the data was performed with the scheduled variation of the experimental conditions such as the movement and rotation of the optical components. In ellipsometry, the optical configurations such as P_R_SA, PSA_R_ and PC_R_SA types should rotate the optical components to change the polarization of the light to obtain enough data to extract *Ψ* and Δ. In LCSI and confocal scanning microscopy, a reference mirror or a specimen needs to be axially scanned. In this case, those mechanical motions of the motorized stage were assumed as uniform, and the motion errors were properly calibrated to reduce the measurement uncertainty.

However, the main drawback of the mechanical scanning is the long-time consumption and unexpected environmental variations such as temperature and vibration. To minimize the environmental disturbances, the electrical devices such as photoelastic [98,99], acousto-optic [100] and liquid crystal [101] components have been used to replace the mechanical devices, which significantly increase the measurement speed.

### 3.2. Spatial Acquisition of the Data

In practical applications, film thickness measurements should be performed at not only a single point but also several points of interest or a field of view area. The fundamental principles of ellipsometry and reflectometry were established at a single point opposed to interferometry, and several research works extended the measured points using imaging optics. In reflectometry, an imaging spectrometer and an area camera were adopted to obtain spectroscopic–spatial images for the line profile of film thicknesses [102,103,104], as shown in Figure 12. Furthermore, a wavelength-swept source was used to obtain the spectral image stack, and 3D film thickness profiles were reconstructed at once [41,42,44,45,46].

Imaging ellipsometry was also proposed to measure 3D film thickness profiles with the combination of ellipsometry and imaging optics. In null type ellipsometry, the 3D film thickness profile was able to be obtained at the area within the depth of field. In case of spectroscopic imaging ellipsometry (SIE), an imaging spectrometer and an area camera are used to obtain the spectroscopic-polarized images. However, previous imaging ellipsometry only focused on the spatial resolution enhancement, and it has moved for measuring the large area of specimen [27,28,29,30,31,32].

### 3.3. Snapshot Measurement

Because the most important issue in film metrology was the confirmation of measurement accuracy and precision, the previous research studies have been focused on developing calibration techniques to reduce the measurement uncertainty caused by the misalignment of optical components, imperfection of optics, environmental variations and calculation errors. On the other hand, the measurement speed has been newly essential for the instrumentation of optical metrology, and it has become a keyword in film metrology.

The desirable scheme for high-speed measurement is snapshot measurement, and several approaches to realize snapshot film measurements have been attempted. In ellipsometry, the temporal polarization variations were replaced with the multi-channeled configuration [105], spatial polarization device based on spatial phase retardation [37]. In addition, a polarization pixelated CMOS camera (PCMOS) was used instead of a rotating analyzer to obtain four different polarized data [32], as shown in Figure 13A. In PCMOS, an individual pixel of the image sensor has its own polarizer, oriented by 0, 45, 90, and 135°, of which a (2 × 2 pixels) unit cell block is repeatedly arranged in the whole imaging area. Therefore, the unit cell of PCMOS is the same as four analyzers with a step of 45°. Subsequently, the PCMOS-based spectroscopic ellipsometry is equivalent to a rotating-analyzer type of spectroscopic ellipsometry, and it can obtain the ellipsometric spectral data for the line at once with a snapshot measurement. The PCMOS was also used in spectroscopic reflectometry [106] as a concept of angle-resolved spectral reflectometry for determination of the thickness of each film layer in multi-layer films, as shown in Figure 13B.

Although the snapshot measurement techniques significantly increase the measurement speed of the system, however, it fundamentally converts the temporal procedure into the spatial one, which causes the loss of measurement accuracy and precision. For example, the use of a PCMOS sacrifices the lateral resolution of the measurement results. In a spatial polarization device, the uniformity of the device affects the measurement results.

## 4. Discussion

In this review, recent publications regarding film metrology during the past two decades were considered, and most of the research works were categorized as the technological point of view. Because of the limited time and effort for this review, a few research studies might be omitted in this review. However, we have carefully searched the publications, which dealt with film metrology in various ways. Although some research works are not mentioned in this review, we believe the technological trend of film metrology was reflected in this review.

In fact, the film metrology has a broad spectrum from plate thickness measurements to single atomic layer measurements. Moreover, various technologies have been proposed and experimentally verified. Therefore, it is difficult to distinguish the capability of each technology for the corresponding valid application. In spite of this ambiguity, the technologies in film metrology were categorized as ellipsometry, reflectometry, confocal microscopy, interferometry and hybrid system in this review, as shown in Table 1. In each technology, of course, several advanced techniques have been developed, and they are summarized.

There are lots of arguments about the criterion between thin and thick film layers in this field, but we decided that the thin film layer should have a thickness of less than 100 nm. This is because confocal microscopy and interferometry did not accurately and precisely determine the film thickness less than 100 nm. In this sense, these two technologies are more appropriate for thick film measurements and even plate thickness measurement. On the other hand, ellipsometry and reflectometry are more suitable for thin film thickness measurements. Especially, ellipsometry has better sensitivity than reflectometry because of the inclined incident light and its sensitive polarization changes on reflection. During the past two decades, most research works have been implemented with this trend. In order to overcome the barrier of each technology, the hybrid systems have been proposed and instrumented. The purpose of hybrid systems was put on the multi-functionality such as the simultaneous measurement of topographic and tomographic film profiles, convenient configurations and better sensitivity for thin film layers. The approaches to combine distinct measuring principles will continue, and more advanced hybrid system will come up.

Another research trend of film metrology is increasing the measurement speed of the system. For the acquisition of the raw data, several snapshot measurement techniques have been proposed and experimentally verified. In spite of relatively low accuracy and precision in current status, the snapshot measurement attempts sufficiently showed the potential of high-speed measurements to replace the current systems. Along with the improvement of current commercialized film metrology system, snapshot measurement system will be further investigated to overcome its current limitations.

Unlike the traditional measurement strategy to analyze the measurement results based on the theory, recently, machine learning (ML) or deep learning algorithms have been applied to film metrology with the huge computing power [107,108,109,110,111,112]. In various science and engineering fields, ML algorithms are being increasingly used to analyze characterization data, although the physical or analytical model is not completely found out. Instead of struggling to figure out the theoretical description, it builds a model based on sample data, so-called ‘training data’, in order to make predictions or decisions without the explicit theory [107]. In film metrology, most of the research works based on the ML algorithm were focused on the analysis of ellipsometric data. In a sense, the optimization techniques used in reflectometry and ellipsometry are regarded as a kind of ML technique, but ML-based film metrology does not need a priori material information. Instead, lots of experimental data sets to train the algorithm should be prepared. Several research works based on ML algorithms have been recently reported, and they measured and monitored the growth of the film layers during the film deposition process [108,109], film thickness variations during the CMP process [110] and the optical property of materials [111]. The ML-based film metrology was recently extended to the determination of a multi-layered wafer stack structure in semiconductor applications [112], as shown in Figure 14, and each layer thickness was able to be non-destructively determined with an average of approximately 1.6 Å as RMS error.

When the target and its film structure are selected in the manufacturing process, machine learning can reduce the calculation time to inspect and measure the target. Therefore, machine learning techniques will be further developed in industrial applications. However, it is not suitable for general purposes because of tremendous data for the training procedure. In case that the innovative training algorithm is developed, machine learning will be further investigated in film metrology.

## 5. Conclusions

In this review, we summarized the recent progress on film metrologies during the last two decades. The technologies were categorized by the fundamental principles, and the measurement schemes are described as data acquisition. The prospect on optical film metrology was carefully provided and discussed with the technology trend.

## Figures and Tables

**Figure 1 micromachines-13-01074-f001:**
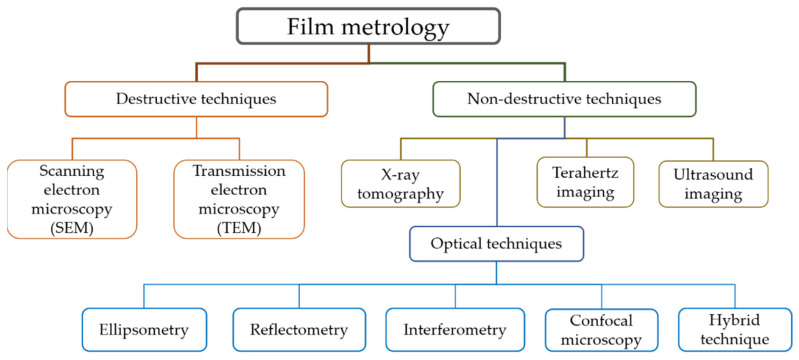
Schematic of technology category in film metrology as the technical point of view.

**Figure 2 micromachines-13-01074-f002:**
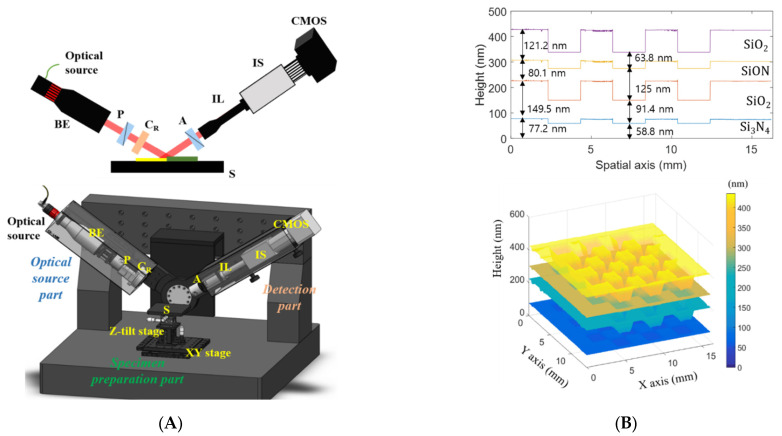
(**A**) Optical configuration of spectroscopic imaging ellipsometry and (**B**) its multi-layered film characterizations originated from Ref. [31]; BE, a beam expander; P, a polarizer; C_R_, a rotating compensator; A, an analyzer; S, a specimen; IL, an imaging lens; IS, an imaging spectrometer.

**Figure 3 micromachines-13-01074-f003:**
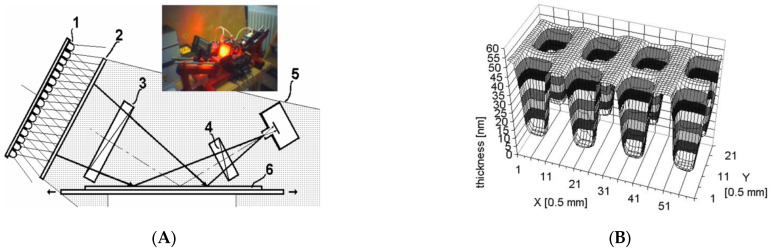
(**A**) Schematic of non-collimated ellipsometry and (**B**) measurement result of a 3D thickness profile originated from Ref. [36]; 1, extended source; 2, diffusor screen; 3, polarizer; 4, analyzer; 5 pinhole and CCD camera; 6, sample with a moving stage.

**Figure 4 micromachines-13-01074-f004:**
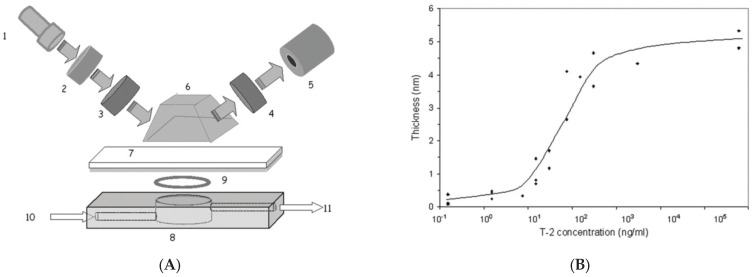
(**A**) Optical configuration of total internal reflection ellipsometry and (**B**) its measurement result for the adsorbed layer thickness on the concentration of T-2 mycotoxin originated from Ref. [24]; 1, the light source; 2, monochromator; 3, polarizer; 4, analyzer; 5, photodetector; 6, glass prism; 7, Cr/Au coated glass slide; 8, cell.

**Figure 5 micromachines-13-01074-f005:**
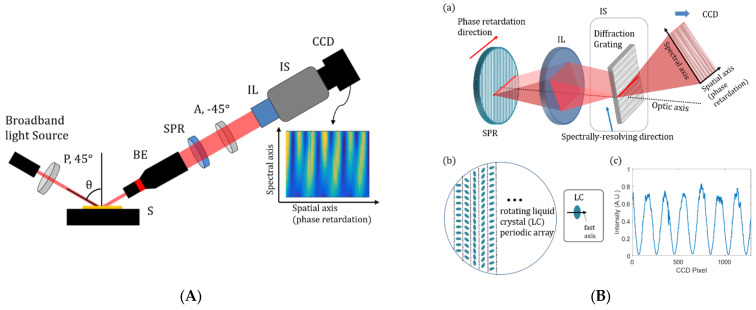
(**A**) Schematic of spatially phase-retarded spectroscopic ellipsometry and (**B**) operating principle of spatial phase retarder (sub-panels: (**a**) acquisition of the single image containing intensity variations corresponding to polarization changes and wavelengths, (**b**) depolarizer based on a rotating LC array as a SPR, and (**c**) intensity variation of linearly polarized light by SPR) (reprinted/adapted with permission from [37] © The Optical Society); P, polarizer; S, specimen; BE, beam expander; SPR, spatial phase retardation plate; A, analyzer; IL, imaging lens; IS, imaging spectrometer.

**Figure 6 micromachines-13-01074-f006:**
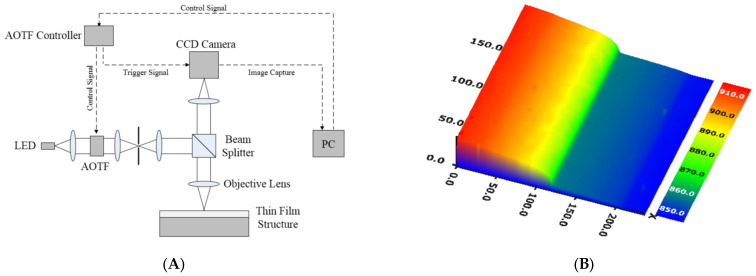
(**A**) Schematic of imaging spectroscopic reflectometry using an acousto-optic tunable filter and (**B**) measurement result of 3D film thickness profile reprinted from Kim et al. *Curr. Opt. Photonics* 2017, 1: 29–33 [40]; AOTF, acousto-optic tunable filter.

**Figure 7 micromachines-13-01074-f007:**
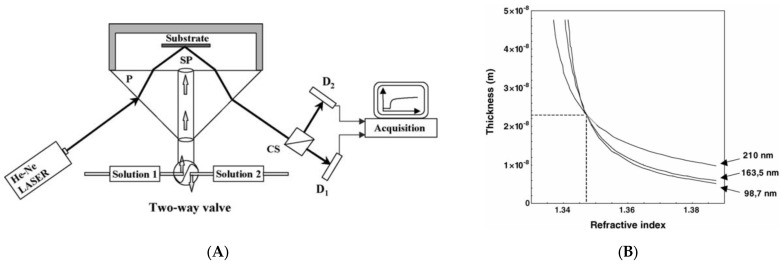
(**A**) Schematic of optical fixed-angle reflectometry and (**B**) graphical determination of mean refractive index and thickness of PVI monolayer originated from Ref. [47]; SP, stagnation point; P, prism; D_1,2_, photodiodes; CS, polarizing beam splitter cube.

**Figure 8 micromachines-13-01074-f008:**
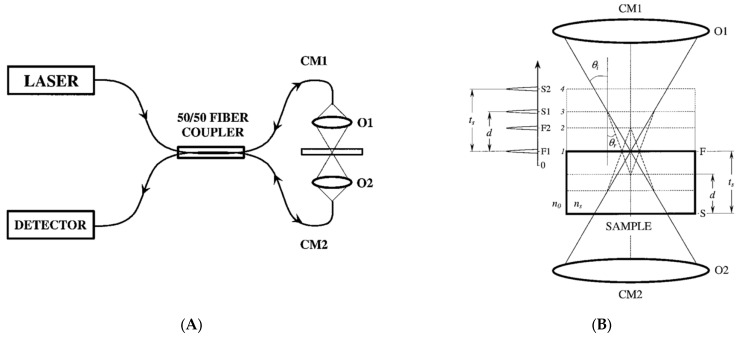
(**A**) Optical configuration of the dual confocal fiber optic system and (**B**) its operating principle by the ray model (reprinted/adapted with permission from [56] © The Optical Society); LASER, He–Ne laser (632.8 nm); O1, O2, confocal focusing objectives; CM1, CM2, first and second confocal microscopes, respectively; DETECTOR, precise digital power meter.

**Figure 9 micromachines-13-01074-f009:**
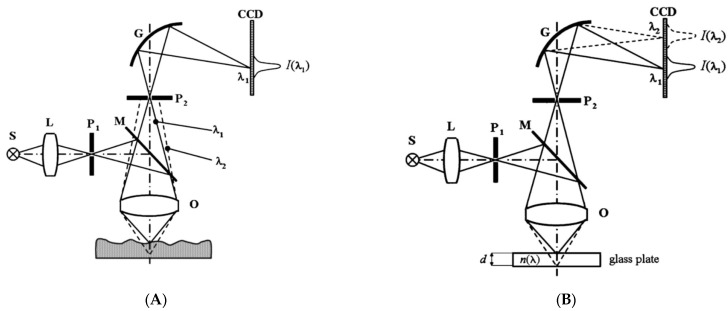
Chromatic confocal microscopy for measuring (**A**) topographic profile and (**B**) a transparent plate thickness (reprinted/adapted with permission from [57] © The Optical Society); S, optical source; L, condenser; P_1,2_, aperture stops; M, beam splitter; O, objective lens; G, grating.

**Figure 10 micromachines-13-01074-f010:**
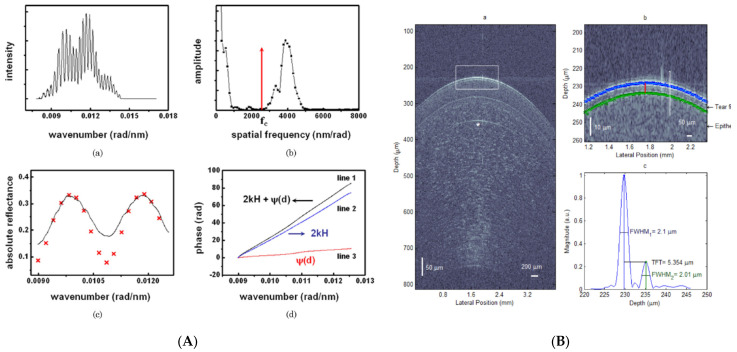
(**A**) An example of optimization procedure to measure film thicknesses in SRI (sub-panels: (**a**) measured intensity of the sample according to the wavenumber from a CCD, (**b**) Fourier-transformed signal of (**a**), (**c**) comparison of measured reflectance of the sample (solid line) and result of reflectance model (x mark), and (**d**) phase distribution maps according to wavenumber (line 1, total phase distribution; line 2, phase distribution due to surface height; line 3, phase distribution due to thin-film thickness)), and (**B**) Fourier peak position detection corresponding to a plate thickness (reprinted/adapted with permission from [75] © The Optical Society) (sub-panels: (**a**) Tomogram of the cornea measured in vivo by the ultrahigh resolution OCT system, (**b**) Close-up view on the tear film around the cornea apex, corresponding to the area delimited by the white rectangle in (**a**), (**c**) OCT periodogram corresponding to the location indicated by the white rectangle in Figure 4B) (reprinted/adapted with permission from [83] © The Optical Society).

**Figure 11 micromachines-13-01074-f011:**
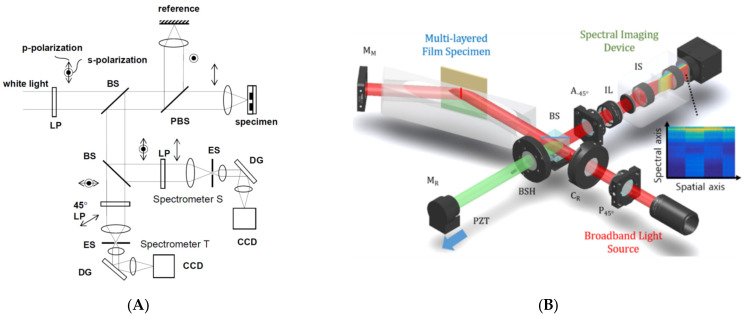
Hybrid film measurement system with (**A**) spectrally-resolved interferometry and reflectometry (reprinted/adapted with permission from [91] © The Optical Society(and (**B**) spectrally resolved interferometry and spectroscopic ellipsometry (reprinted/adapted with permission from [95] © The Optical Society); LP, linear polarizer; BS, beam splitter; PBS, polarizing beam splitter; ES, entrance slit; DG, dispersive grating; CCD, charge coupled device; P, polarizer; C_R_, rotating compensator; BS, beam splitter; BSH, beam shutter; A, analyzer; IL imaging lens; IS, imaging spectrometer; M_R_ and M_M_, a reference and measurement mirrors; PZT, piezoelectric transducer.

**Figure 12 micromachines-13-01074-f012:**
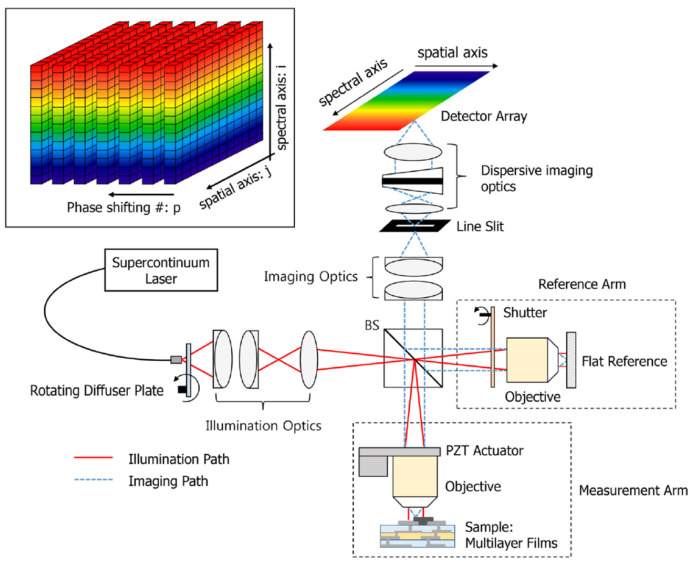
Imaging spectroscopic reflectometry combined with SRI originated from Ref. [104].

**Figure 13 micromachines-13-01074-f013:**
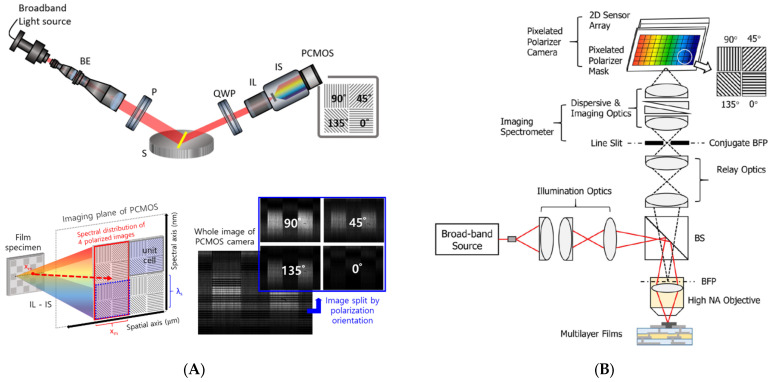
Schematic of (**A**) PCMOS-based spectroscopic ellipsometer (reprinted/adapted with permission from [32] © The Optical Society) and (**B**) PCMOS-based spectroscopic reflectometry (reprinted/adapted with permission from [106] © The Optical Society); BE, beam expander; P, 45° rotated polarizer; S, specimen; QWP, 45° rotated quarter-wave plate; IL, imaging lens; IS, imaging spectrometer; PCMOS, polarization-pixelated CMOS camera.

**Figure 14 micromachines-13-01074-f014:**
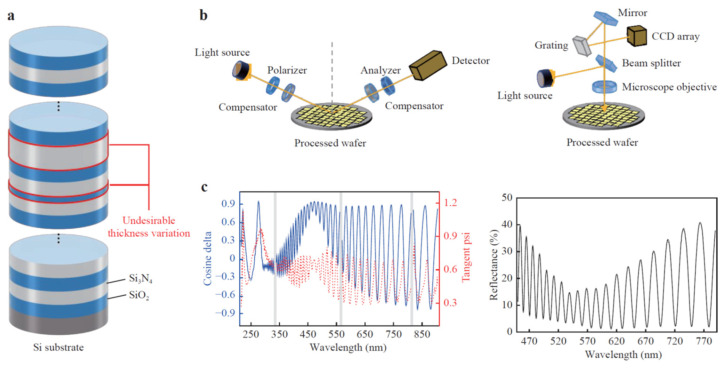
Machine learning strategy for measuring multi-layered wafer stack originated (sub-panels: (**a**) multi-layer structure with alternating silicon oxide (blue) and silicon nitride (white) layers on a Si substrate, (**b**) Schematics of a typical spectroscopic ellipsometer (**left**) and reflectometer (**right**), (**c**) Examples of ellipsometric (**left**) and reflectance (**right**) measurement data for machine learning) from Ref. [112].

**Table 1 micromachines-13-01074-t001:** Summary of the comparison between film measurement techniques.

	Ellipsometry	Reflectometry	ConfocalMicroscopy	Interferometry	Hybrid Technique
Thin film measurement	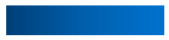	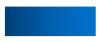	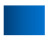	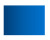	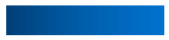
Thick film (plate) measurement	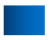	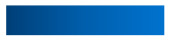	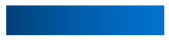	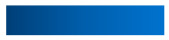	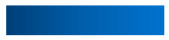
Topographic measurement			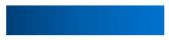	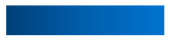	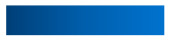
Measurement speed	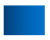	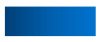	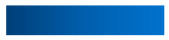	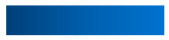	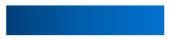
Precision	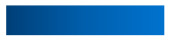	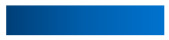	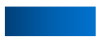	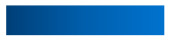	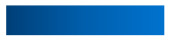
Research trend	Snapshot measurement	Thin film measurement	Motionless measurement	Theoretical model-based algorithm	Multi-functional measurement

The length of the blue bar (
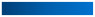
) indicates the relative performance value.

## Data Availability

Not applicable.

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
