# Peer review of "Recent Progress on Optical Tomographic Technology for Measurements and Inspections of Film Structures"

_micromachines, 2022, doi:10.3390/mi13071074_

Round 1

Reviewer 1 Report

"Recent progress on optical tomographic technology for measurements and inspections of film structures"  is a comprehensive review in field of thin/thick film metrology. It reviews the different NDT techniques, such as reflectometry, ellipsometry, confocal microscopy, low coherent and dispersive interferometry and hybrid methods in the context of film characterization.

The review describes the advantages and limits of the different optical methods for the interested reader and provides a good survey in general.

Some of the optical schemes within the figures might be improved by including the abbreviations/symbols in the caption of the figures.

For the reproducing of schemes within the figures, the copyrights are given by the corresponding journals?

Author Response

"Recent progress on optical tomographic technology for measurements and inspections of film structures" is a comprehensive review in field of thin/thick film metrology. It reviews the different NDT techniques, such as reflectometry, ellipsometry, confocal microscopy, low coherent and dispersive interferometry and hybrid methods in the context of film characterization.

Answer: Thank you for the summary of this manuscript.

The review describes the advantages and limits of the different optical methods for the interested reader and provides a good survey in general.

Some of the optical schemes within the figures might be improved by including the abbreviations/symbols in the caption of the figures.

Answer: Thank you for the comment. We added the full names of abbreviations in the captions as suggested.

For the reproducing of schemes within the figures, the copyrights are given by the corresponding journals?

Answer: Thank you for the comment. We obtained the permissions of figures reproduced in the manuscript.

Reviewer 2 Report

This review is devoted to recent progress in Optical Tomographic Technology for Measurements and Inspections of Film Structures. This work is in the scope of the Micromachines (with a scope of science and technology of small structures, e.g. film structures). However, I have some important comments that should be addressed prior to acceptance.

Major:

I. The structure of the sections is confusing. Section 2 looks as related to physical principles (such as Confocal microscopy, Interferometry, Reflectometry...). But Subsection 2.6 related to machine learning (ML). ML is not physical principle but more computer science approach (data and signal processing approach). It looks like the Section related to signal and data processing approaches are also required in this sense.

II. The Discussion and Conclusion section looks too short and too descriptive. There are lack of systematization and visualization of the previous sections. As for summarizing this review I suggest to the authors to include the systematization table in the Discussion section to emphasize the pros and cons of each approach as well as some schematic illustration of the proper application (or cases) for each approaches.

Minor:

1. Some figures, such as Fig.10 contains several panels. However, the figure caption not disclose these panels. Here also I can note that the only two notations (a) and (b) are insufficient. There are 7 panels and it should be (a), (b), (c), (d) (e) (f) (g) or (a-1), (a-2), (a-3), (a-4), (b-1) , (b-2) , (b-3) to improve readability.

2. Figure 12 is absolutely unclear and not described in this review. As it seen it was taken from [103]. However, the pipeline of this ML strategy is unclear. Please indicate the details in the manuscript text.

2.1. Due to the review status of this manuscript it also should be emphasized in details how and where the various types of ML help to improve the measurements and inspections of film structures. Also this section should be better structured in order to achieve coherence between subtopics. For example, various ML techniques to improve various physical approaches can be indicated (for OCT, as Confocal microscopy, Interferometry, Reflectometry, LSI, LSCI etc.).

3. There are a lot of OCT refs. As well as Fig.10(b) shows OCT image. However, in the text the OCT (optical coherence tomography) notation is not used. It is covered by some other terminology. Maybe it should be also added where really OCT were used.

Author Response

This review is devoted to recent progress in Optical Tomographic Technology for Measurements and Inspections of Film Structures. This work is in the scope of the Micromachines (with a scope of science and technology of small structures, e.g. film structures). However, I have some important comments that should be addressed prior to acceptance.

Answer: Thank you for the comment. We carefully considered the comments as follows.

Major:

  1. The structure of the sections is confusing. Section 2 looks as related to physical principles (such as Confocal microscopy, Interferometry, Reflectometry...). But Subsection 2.6 related to machine learning (ML). ML is not physical principle but more computer science approach (data and signal processing approach). It looks like the Section related to signal and data processing approaches are also required in this sense.

Answer: Thank you for the comment. As suggested, we move ‘ML’ section from section 2 to section 4 (Discussion), and ML was carefully considered for applying in film metrology. We think it was well structured.

  1. The Discussion and Conclusion section looks too short and too descriptive. There are lack of systematization and visualization of the previous sections. As for summarizing this review I suggest to the authors to include the systematization table in the Discussion section to emphasize the pros and cons of each approach as well as some schematic illustration of the proper application (or cases) for each approaches.

Answer: Thank you for the comment. As suggested, we summarized the methods as a summary table.

Minor:

  1. Some figures, such as Fig.10 contains several panels. However, the figure caption not disclose these panels. Here also I can note that the only two notations (a) and (b) are insufficient. There are 7 panels and it should be (a), (b), (c), (d) (e) (f) (g) or (a-1), (a-2), (a-3), (a-4), (b-1) , (b-2) , (b-3) to improve readability.

Answer: Thank you for the comment. As suggested, the descriptions for sub-panels in all figures were included.

  1. Figure 12 is absolutely unclear and not described in this review. As it seen it was taken from [103]. However, the pipeline of this ML strategy is unclear. Please indicate the details in the manuscript text.

Answer: Thank you for the comment. With the aids of descriptions of sub-panels of the figure, the figure was able to be clear. We think more detailed information can be found from its reference.

2.1. Due to the review status of this manuscript it also should be emphasized in details how and where the various types of ML help to improve the measurements and inspections of film structures. Also this section should be better structured in order to achieve coherence between subtopics. For example, various ML techniques to improve various physical approaches can be indicated (for OCT, as Confocal microscopy, Interferometry, Reflectometry, LSI, LSCI etc.).

Answer: Thank you for the comment. We move ‘ML’ section from section 2 to section 4 (Discussion), and ML was carefully considered for applying in film metrology.

  1. There are a lot of OCT refs. As well as Fig.10(b) shows OCT image. However, in the text the OCT (optical coherence tomography) notation is not used. It is covered by some other terminology. Maybe it should be also added where really OCT were used.

Answer: Thank you for the comment. As suggested, OCT was mentioned in this review.

“By adopting interferometric principles, optical coherence tomography (OCT) has been increasingly developed and commercialized in biomedical area,”

However, the technical terminologies used in this review are determined as a principle-based point of view for the unity and coherence of this review because lots of various technologies are introduced. For example, a ‘time-domain OCT (TD-OCT)’ is expressed as a ‘low coherence scanning interferometry (LCSI)’ and so on.

Round 2

Reviewer 2 Report

The authors have made a great effort to address all my comments. This review article can be accepted in present form.